# Universal Screening for CCHD in Saudi Arabia: The Road to a ‘State of the Art’ Program

**DOI:** 10.3390/ijns6010013

**Published:** 2020-02-24

**Authors:** Fahad AlAql, Huda Khaleel, Vetha Peter

**Affiliations:** 1Neonatal Services Improvement Program, Ministry of Health (MOH), Riyadh 11525, Saudi Arabia; jeba_77@yahoo.co.in; 2Pediatric Cardiology Department, Heart Health Center, KSMC, Riyadh 12748, Saudi Arabia; hkhaleel@ksmc.med.sa

**Keywords:** Saudi Arabia, CCHD, Congenital Heart Disease, national project, Pulse Oximetry Screening

## Abstract

Critical congenital heart disease (CCHD) has been defined as structural heart defects that are usually associated with hypoxia in the newborn period and have potential for significant morbidity and mortality early in life. CCHD has been estimated to be present in ∼3 in 1000 live births, including Saudi Arabia. Pulse Oximetry Screening (POS) is a highly specific and moderately sensitive test for detecting CCHD with very low false-positive rates. The Kingdom of Saudi Arabia is among high-income countries with a population of more than 33 million and more than 600,000 annual live births. In 2015, the Universal Screening Program for CCHD using Pulse Oximetry was approved in Saudi Arabia. It is expected that any new national program will undergo a learning curve and face many challenges. We believe that developing countries may face different challenges during implementation of such national projects, but the success achieved by Saudi Arabia in implementing the program was mainly due to good preparation before launching the project and advancements in the use of the technology involved in this project. Since starting the universal CCHD screening in 2016, more than 900,000 babies have been screened in Saudi Arabia and many lives have been saved using this safe, non-invasive, inexpensive, and reasonably sensitive test.

## 1. Introduction

Congenital Heart Disease (CHD) refers to structural disorders of the heart and great vessels that are present at birth. CHD affects 8 per 1000 live births and is the most common type of birth defect worldwide, including Saudi Arabia [1,2,3,4]. Globally, CHD affects over one million live births annually, and accounts for thirty percent of deaths in children with birth defects [3,5].

Critical congenital heart disease (CCHD) is a term that refers to a group of serious heart defects that are present from birth. These abnormalities result from problems with the formation of one or more parts of the heart during the early stages of embryonic development. CCHD prevents the heart from pumping blood effectively or reduces the amount of oxygen in the blood. As a result, organs and tissues throughout the body do not receive enough oxygen, which can lead to organ damage and life-threatening complications [6,7]. Individuals with CCHD usually require medical or/and surgical intervention soon after birth.

Although babies with CCHD may appear healthy for the first few hours or days of life, signs and symptoms soon become apparent. Babies sent home with an undetected heart defect are at risk of serious complications within the first few days or weeks of life that may require emergency care and may even be fatal. POS is an evidence-based tool that can identify most of these babies so they can receive prompt care and treatment [7].

## 2. CCHD Screening

Pulse oximetry screening (POS) for CCHD has been shown to be a simple reliable method used to determine arterial oxygen saturation of hemoglobin in arterial blood and has been widely used in many areas of clinical medicine [7]. Particularly useful for identifying asymptomatic infants with CCHD in well-baby units, the importance of this screen is in its ability to allow for the early detection of CCHD prior to the infant being discharged from the hospital [7,8]. Late detection of CCHD, after hospital discharge, has been shown to increase morbidity and mortality. Furthermore, earlier diagnosis means earlier treatment, which means a better outcome, in addition to fewer financial and other costs [9].

The Kingdom of Saudi Arabia is among the high-income countries with a population of more than 33 million and more than 600,000 annual live births [10]. Saudi Arabia has a well-developed healthcare system providing a full range of health care services. Maternity and neonatal care is provided within a network of more than 450 hospitals divided into different levels of care delivery. Of these, more than 90 hospitals are able to provide the full range of service specialties and subspecialties [10].

## 3. Establishing Neonatal CCHD Screening in the Kingdom

In 2012, the Saudi Ministry of Health (MoH) established a neonatal services improvement program to supervise, monitor and lead changes in neonatal care in the country. This program was fully funded and supported by the government. Furthermore, to ensure full coordination between the various departments in decision-making, a neonatal advisory committee was formed by the Ministry of Health. The committee members include doctors, nurses, representatives from health education, medical consumables and medical equipment departments. Since 2012, a neonatal services improvement program has been continuously working to support neonatal services in the country through more than 40 improvement projects, one of which is Universal CCHD Screening program.

In 2014, the Universal Screening program for CCHD in Saudi Arabia was thoroughly discussed during several neonatal advisory committee meetings; the discussion included evidence-based studies, and international and local experiences. In 2015, the neonatal advisory committee approved the project and submitted their recommendations; first, that universal screening for CCHD using Pulse Oximetry should be routinely performed on all healthy newborns to enhance the early detection of critical congenital heart disease. Second, that optimal screening for CCHD disease should include a prenatal ultrasound, newborn physical examination and pulse oximetry screening. Third, that CCHD screening should be performed on the right hand and either foot. Lastly, that screening for CCHD should be performed for all newborns between 12 to 24 h of age.

The committee recommendations were approved by the MoH and all hospitals were requested to start the implementation of CCHD screening in collaboration with the neonatal services improvement program, as the first and only country in the Middle East at that time. 

## 4. The Initiation of CCHD Screening in the Kingdom of Saudi Arabia 

To ensure a successful national program with an optimal implementation rate, several steps have been taken by the neonatal services improvement program, such as:

### 4.1. Creating a National Steering Committee

The national steering committee includes neonatologists, pediatric cardiologists and nurses. The committee conducts meetings every two months with main tasks including the supervision of the overall project and ensuring the comprehensiveness and quality of the service, a periodical database review to ensure appropriate services are provided to all positive case newborns, and to ensure proper education for both hospital staff and parents about CCHD screening.

### 4.2. Formulating Local Teams in Each Hospital

Each hospital was requested to form their own local team that would consist of one pediatrician, one pediatric cardiologist (if available) and one to three nurses, depending on the capacity of the hospital. The local team is responsible for the supervision of the project in their hospital; they must ensure service quality, training for the staff, education of the parents, and communication with the national steering committee.

### 4.3. Continuous Training and Workshops for the Nursing Staff

A two-day structured competency training was established for the nurses who would perform the newborn screening, including CCHD screening and entering of the results into the national CCHD screening registry. A certificate of “super-user” was given to those nurses who were competent to perform the screening. These nurses were also eligible to train other nurses on how to perform the CCHD screening. 

### 4.4. Formulation of CCHD Screening National Policies and Procedures

National CCHD screening policies and procedures were formulated and approved by the neonatal services improvement program in Saudi Arabia before being disseminated to all heath care facilities in the country. The approved policies and procedures are comprehensive, simple, and include steps on how to perform the CCHD screen. 

### 4.5. Using (Smart) Pulse Oximetry Screening (POS) Machines

To reduce human error, only (smart) Pulse Oximetry Screening (POS) machines, with an integrated CCHD screening protocol, were used. These machines contain screening protocol instructions and integrated guidance, which can be easily followed by the operators; negative results are shown in green and positive results are shown in red.

### 4.6. National CCHD Screening Registry

The MoH has created a national secure webpage system, with secure servers centrally located in the MoH, which track and save the results of the screenings of all newborns. To empower the national registry and to reduce error, the CCHD screening protocol was integrated in the national registry. The registry system was integrated with the national birth notification system to effectively obtain full reports. This high caliber registry system is an ideal means of providing the decision-makers with live dashboards to monitor the progress, compliance and screening results, as well as providing researchers with all the necessary data.

### 4.7. Integration between the Screening Devices and the National Registry System

To eliminate human error during data entry and to conserve the time taken to perform the screening, we have developed a unique worldwide integration system that connects screening devices with the national registry system. This allows the screening result to be retrieved automatically from medical devices without any human intervention.

### 4.8. Reporting System (Feedback to the Hospitals)

Each hospital receives detailed monthly report about their CCHD screening performance. This may encourage the local teams to carry out prompt and accurate corrective actions if necessary. Prompt and accurate feedback is essential to the success of the screening program.

### 4.9. Implementation Phases

The CCHD screening program in Saudi Arabia was initiated in phases. The implementation phase of the screening began on 1 January 2016 in the largest 30 hospitals, a further 70 hospitals in 2017, and the remaining hospitals followed in January 2018. 

### 4.10. Follow-up with Parents

Parents are notified about the CCHD screening results by the screening team; furthermore, the result is documented in the baby’s medical record and in the vaccination card (Figure 1). The national registry system has the capability to generate a screening report for the parents if requested. 

### 4.11. Workshops for General Pediatricians on how to Use the Echocardiogram Machine

Due to the unavailability of an echocardiogram facility in many local hospitals, short trainings and workshops were conducted for general pediatricians on how to use an echocardiogram machine and perform CCHD screening with the remote help of our pediatric cardiologists. This experience was very helpful and reassuring for both the general pediatricians and the parents. 

## 5. Challenges

Despite good preparation before and during implementation of the CCHD screening, it is expected that any new national program will undergo a learning curve and face many challenges. The main challenges for our CCHD screening program were identified as follows: first, dealing with frequent changes in the local screening team, as the trained nurses involved in CCHD screening are frequently changing, and this may affect the performance and success of the program. Second, improving data entry and reducing human error. Third, close monitoring of the test-positive cases. Last, close monitoring of false negative cases. 

The steering committee is taking all possible action to resolve the issues. They are conducting continuous training for nurses to become super-users—competent nurses who are eligible to train other nurses on how to perform CCHD screening. With regards to the collection of data and reducing human error, automation of the process with frequent updates and a regularly upgraded registry system will help in reducing error and the time needed for screening. Newborns with positive results should undergo a thorough evaluation by the most responsible health care provider on duty to evaluate the baby and to decide if referral to a pediatric cardiologist is needed. If a pediatric cardiology service is not available at the local hospital, an urgent referral to the appropriate center should be made. False negative results are a real challenge, but we believe good communication between the different cardiac centers and the national steering committee can help in discovering these cases. 

## 6. Conclusions 

Pulse Oximetry Screening (POS) is a highly specific and moderately sensitive test for detecting CCHD with very low false-positive rates [7,11]. Current evidence supports the introduction of routine screening for CCHD in asymptomatic newborns before discharge from the well-baby nursery [7].

We believe that developing countries may face different challenges during the implementation of such national projects. We believe that the success achieved by Saudi Arabia in implementing the program was due to good preparation before launching the project and advancements in the use of the technology involved in this project.

Since starting the universal CCHD screening in 2016, more than 900,000 babies have been screened in Saudi Arabia, with 0.26% being positive screening cases, which is in keeping with published global studies [12,13]. Many positive referral cases were successfully treated early in the different cardiac centers in Saudi Arabia, and due to early detection by CCHD screening, many lives have been saved using this safe, non-invasive, inexpensive, and reasonably sensitive test. 

## Figures and Tables

**Figure 1 IJNS-06-00013-f001:**
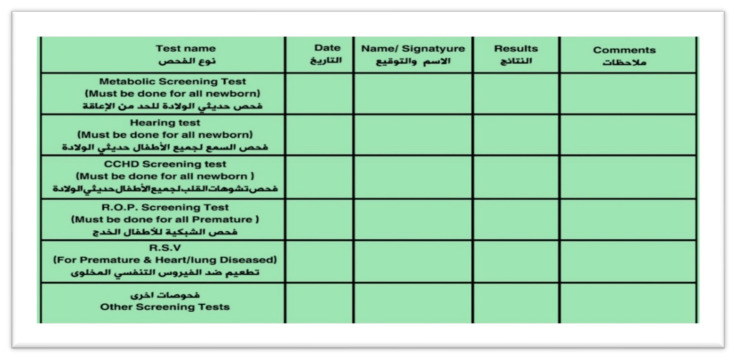
Documenting CCHD screening result in the baby’s vaccination card.

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
