# Peer review of "Universal Screening for CCHD in Saudi Arabia: The Road to a ‘State of the Art’ Program"

_2409-515X, 2020, doi:10.3390/ijns6010013_

Round 1

Reviewer 1 Report

The authors have to be congratulated for this ambitious and well structured project. The manuscript is of interest to all involved in perinatal / neonatal care. The experiences reported are of special value to those beginning to implement CCHD screening programs. The amount of data with precise two site saturation readings of over 900.000 newborns exceeds those in any published report so far. Complete analysis might help to resolve some the remaining open questions concerning POS (i.e. optimal screening algorithm, optimal timing of screening).

Author Response

Thanks for this feedback, we are working to participate and contribute to the research of this field.

Based on the other reviewer's suggestions, kindly see the updated file.

Thanks and best regards  

Reviewer 2 Report

Thank you for asking me to review this paper

I have some comments which if addressed will enhance the paper.

Abstract 

i) It would be helpful to define CCHD and describe the incidence of this rather than CHD which is not the target of screening.

ii) It would be useful to briefly describe the screening test - i.e. pulse oximetry (which is not currently mentioned).

Introduction

Line 34 - infants usually require surgical intervention Medical intervention is just for stabilisation.

Line 36/37 - signs and symptoms are often absent! this is why screening is so important. this must be emphasised.

The first 2 references are not entirely appropriate - it would be better to have peer-reviewed published papers rather than weblinks.

Refs 3 and 4 are the same. It would be useful to include other systematic reviews as well (e.g. Thangaratinam Lancet 2012)

It would be useful to have some background into the research of pulse oximetry screening

Line 48 Earlier diagnosis means better outcome not just financial cost savings. This should be acknowledged and referenced.

Line 102 'Smart' should be expanded and explained

I'm not sure the green card figure adds anything I would remove.

Likewise the map has no legend and I'm not sure what it contributes

Author Response

i) & ii) we can change the abstract as the following: (Critical congenital heart disease (CCHD) has been defined as structural heart defects that are usually associated with hypoxia in the newborn period and have potential for significant morbidity and mortality early in life. CCHD has been estimated to be present in ∼3 in 1000 live births, including Saudi Arabia.) (Pulse Oximetry Screening (POS) is a highly specific and moderately sensitive test for detecting CCHD with very low false-positive rates.) please see revised Abstract 

Line 34) stabilization is an important step in the management, and to make it more clear we can change it to (... require medical or/and surgical intervention soon after birth.)

Line 36/37) changes done in the paragraph to be more clear 

The first 2 references are not entirely appropriate) Refs 3 and 4 are the same): please see the revised references 

background into the research of pulse oximetry screening: was integrated into the (CCHD SCREENING) section

Line 48 Earlier diagnosis means better outcome) agree, kindly see changes and references

Line 102 'Smart'): I think it is clearly mentioned, some changes are done, I hope it is more clear   

map ): is used to indicate that Saudi is the first middle east country started CCHD screening, changes were done to make it more clear 

green card): to indicate that the result is communicated with the parents using the vaccination card, I think it is good for developing countries  

Round 2

Reviewer 2 Report

My comments have been addressed - thank you

Guest Editor comments

This is a wonderful overview of CCHD screening implementation and provides information regarding the initiation of a sustainable successful national program.  Minor edits to language and clarification in some sections to indicate whether actions are part of future plans or the current program would be helpful.  For example, "each hospital will receive" (receives?) and "parents will be notified" (are notified?).  A little bit more analysis of their screening outcomes would also strengthen this manuscript.  Explicitly state the numbers not just the percentage or "with 0.26% being positive screening cases, which is a worldwide percentage" means...?  It would be great if the team could break down their outcomes - how many were true positives for CCHD, positives for other important conditions like CHD or neonatal diagnosis like PNE or sepsis.  I think what they are saying is that the test positive rate is in keeping with other published global studies but it would be better to spell it out and cite for example they could use (Thangaratinam Lancet 2012).

Author Response

The suggested modifications and the language corrections all done

Regarding the 0.26% of tested-positive cases citation was added, but we don’t have a detailed analysis of the cases at this time, and we are planning for full analysis, but it will take time to finish it, and we can't add it at this paper.